# Utilizing Evolution Strategies to Train Transformers in Reinforcement Learning

## Abstract

We explore the capability of evolution strategies to train an agent with a policy based on a transformer architecture in a reinforcement learning setting. We performed experiments using OpenAI's highly parallelizable evolution strategy to train Decision Transformer in the MuJoCo Humanoid locomotion environment and in the environment of Atari games, testing the ability of this black-box optimization technique to train even such relatively large and complicated models (compared to those previously tested in the literature). The examined evolution strategy proved to be, in general, capable of achieving strong results and managed to produce high-performing agents, showcasing evolution's ability to tackle the training of even such complex models.

## 1 Introduction

The problem of reinforcement learning is considered one of the most difficult in the field of machine learning. There are many approaches to solving it (Sutton & Barto, 2018). Some are based on computing a gradient to optimize the objective, some are derivative-free. One such class of general derivative-free optimization algorithms are evolutionary algorithms (De Jong, 2016). Their subclass of evolution strategies (Rechenberg, 1973) has been proved to be a viable alternative to gradient approaches for the (deep) reinforcement learning (Pagliuca et al., 2020). Although the gradient approaches generally have better sample utilization, the evolution strategies are greatly parallelizable. Moreover, evolution strategies have better exploration of possible solutions and the agents trained using these methods are usually more diverse than those trained by the gradient-based algorithms. They can even incorporate techniques that vastly improve the exploration even more, such as searching for novelty instead of, or in addition to just seeking better performance. This yields novelty search (Lehman & Stanley, 2011a;b) or quality-diversity (Pugh et al., 2016) algorithms. An example of such algorithm for reinforcement learning, that is fairly simple, yet highly efficient, is OpenAI-ES (Salimans et al., 2017).

Transformer architecture (Vaswani et al., 2017) is lately the go-to solution in the field of neural networks and supervised learning for an ever-growing range of problems. And recently, there have been attempts to reformulate reinforcement learning as a sequence modeling problem and to leverage the capabilities of the transformers in such tasks to obtain a new approach for solving this class of problems, yielding models such as Decision Transformer (Chen et al., 2021) or Trajectory Transformer (Janner et al., 2021). The Decision Transformer was originally introduced as a model for offline reinforcement learning using a supervised learning of sequence prediction, but the authors claim it would function well even in the classical reinforcement learning tasks.

We decided to subject the combination of the evolution strategies and the Decision Transformers to experiments, and test the ability of derivative-free algorithms to train this more complicated and bigger – compared to the simple feedforward models that had been experimented with in the literature so far – transformer architecture. It might be the case, that for large and complicated models it may be hard, or even almost impossible, to be trained from scratch using an evolutionary approach. Therefore, we wanted to experiment with first pretraining the agent using a supervised learning of sequence prediction on data generated by a smaller, potentially weaker model, which can be trained using any arbitrary reinforcement learning method. We are not searching for use cases where we gain an edge over existing approaches by using Decision Transformers trained via evolution; rather we aim to show that evolution works well even on complex models, and thus we need not hesitate to use

the complex models with, e.g., evolutionary reinforcement learning algorithms (Sigaud, 2023; Li et al., 2024), the hybrid state-of-the-art reinforcement learning algorithms that combine gradients and evolution, performing better than the two individually. Furthermore, the ability of evolution to train the Decision Transformers would also give us an easy-to-adapt algorithm to train the Decision Transformer in a fully online regime, as compared to the original offline training (Chen et al., 2021) or the offline-to-online regime of Online Decision Transformer, a slight Decision Transformer modification (Zheng et al., 2022).

The main contribution of this paper is demonstrating that evolution strategies can scale to transformer-based reinforcement learning agents, showing that even a simple algorithm such as OpenAI-ES is capable of training large sequence models like the Decision Transformer. We provide insights into the interaction between gradient-based pretraining and evolutionary optimization, explaining why pretrained models may hinder evolutionary training and highlighting the inherent robustness of evolution-trained agents. With experiments in MuJoCo (Todorov et al., 2012) Humanoid and Atari (Bellemare et al., 2013), we empirically validate the robustness and versatility of evolution strategies across both continuous-control and high-dimensional visual domains.

## 2 BACKGROUND

### 2.1 EVOLUTION STRATEGIES

Evolutionary algorithms are a large and quite a successful family of black-box derivative-free optimization algorithms. One subclass of such nature-inspired optimization methods are *evolution strategies*. Introduced as a tool for dealing with high-dimensional continuous-valued domains (Rechenberg, 1973), evolution strategies work with a population of individuals (real-valued vectors). In each generation (iteration / population in the iteration), they derive a new set of individuals by somehow mutating (perturbating) the original population; the new set is then evaluated (with respect to a given objective function), and a new generation is formed based on these new evaluated individuals taking into account their fitness (objective function value).

In order to use an evolution strategy as a reinforcement learning algorithm, we use agents represented by a neural network, or, more specifically, by a real-valued vector of the network weights, as the individuals for the algorithm. Then the only thing needed is to set the fitness of each individual as the mean return, the mean cumulative reward of the individual agent from several episodes. Various evolution strategy algorithms were proposed for this purpose (Pagliuca et al., 2020).

In this paper, we will be working with *OpenAI-ES* (Salimans et al., 2017). The population is represented by a distribution over the agent's (neural network's) parameters, the distribution being a Gaussian, whose mean value is our current solution to the given problem and from which the offsprings are sampled each generation. These are then evaluated and their evaluation is used to update the parameters (in our case only the mean value) of the distribution so that we obtain a higher expected fitness for the future samples. This update is performed using an approximation of natural gradient. In our case, when we derive the parameter updates from the Gaussian distribution with the same variance for each parameter, as demonstrated in a paper by Pierrot et al. (2021), it is obtained by renormalizing (rescaling) the update with respect to uncertainty, in other words dividing by the variance. The algorithm is also designed in such a way that it is highly parallelizable with interprocess communication kept at bare minimum. For more details or for a discussion of the design choices, we refer our readers to the original paper (Salimans et al., 2017).

### 2.2 TRANSFORMERS

*Transformers* have surged as a new sequence-to-sequence architecture (Vaswani et al., 2017), as an alternative to recurrent neural networks. Since then they have yielded great results in natural language processing (NLP) tasks, fueling even the current surge of chatbots, and they even got adapted, e.g., to image recognition (Dosovitskiy et al., 2021). As a rule of thumb, they seem to have a great generalization power; the greater the larger the model employed. However, they also require a lot of training data to achieve these great results.

The transformers interlace classical feedforward layers with self-attention layers. And it is the self-attention to which the transformers owe their success. For each element of an input sequence,

the self-attention constructs a "key", a "query", and a "value" using fully connected layers of neurons. Then, to compute an $i$-th element of an output sequence, it takes a combination of all the values, each weighted proportionally to the product of the query belonging to the given ($i$-th) position and a key corresponding to the value – hence combining the information from the whole input sequence to produce every single element of the output sequence, as shown by the following equation.

$$output_i = \sum_{j=1}^{n} \text{softmax} \left( query_i^T \cdot all\_keys \right)_j \cdot value_j$$

We can use a mask and for every element hide the part of the sequence that is behind the element, and so use just the information that came before in the sequence to derive the output. This is called a causal masking, and such a transformer that uses this masking we then call a causal transformer.

In the problem of reinforcement learning, we want the agent to choose actions at individual timesteps, such that it maximizes its return. This can be, however, viewed as a sequence modeling problem. Thus, the *Decision Transformer* was introduced (Chen et al., 2021).

Its main idea is that we want the agent's policy to produce an action based on the whole history (or the part which squeezes into the context window) of past observations and undertaken actions. And to have some way to affect the agent's performance, we add a conditioning on a *return-to-go*, which is a return we want to obtain from a given step until the end of the episode.

The Decision Transformer consists of a causal transformer; embeddings for returns-to-go, observations (states of the environment), and actions; position encoder; and a linear decoder to transform the output of the transformer into actions, as shown in Figure 1.

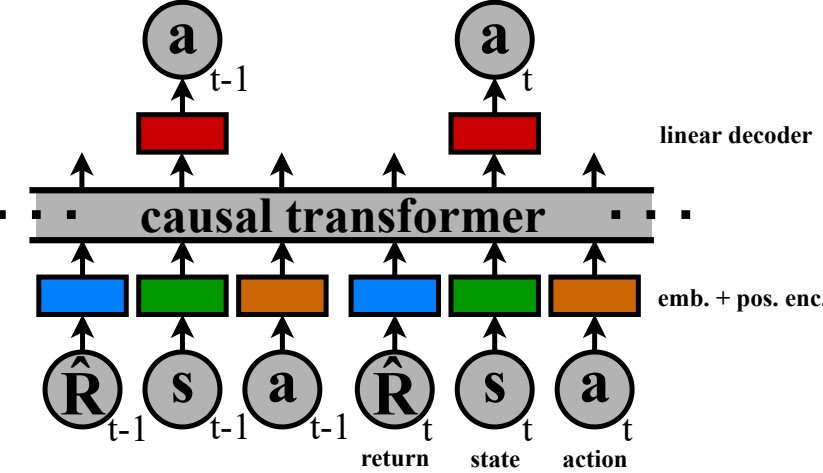

Figure 1: Decision Transformer architecture (Chen et al., 2021)

Every timestep, we feed the model with a sequence of past triplets return-to-go, observation, and action performed, adding the current return-to-go and observation (and a placeholder for the yet unperformed action). They get embedded by their respective embeddings, the positional encoding is added, and everything is then fed through the transformer. We then decode the last output state of the transformer to obtain the action to be carried out. Every part of the timestep triplet – return-to-go, observation, and action belonging to the same timestep – shares one positional encoding, as opposed to the classical transformer, where every input sequence element gets its own. Returns-to-go are constructed in a recursive manner. The user has to supply the original one for the first timestep (the desired performance of the model, in other words target return), while for all the following timesteps the return-to-go is constructed by subtracting the last reward obtained from the return-to-go belonging to the previous timestep.

## 2.3 OUR OPENAI-ES IMPLEMENTATION

Mainly due to different computational resources available to us, as well as the need to interchange the models and the environments, we decided to create our own implementation of OpenAI-ES.[1] Yet, in the process of creating this implementation we noticed some inconsistencies between the original paper introducing the algorithm (Salimans et al., 2017) and their provided code. We decided to adhere more to the paper when these dissimilarities occured and here we state our reasoning for the two most significant differences.

First and foremost, in the paper (Salimans et al., 2017), the use of a weight decay is mentioned as a form of keeping the effect of newly added updates still significant enough. After every update, we decay the weights of our model, which is the distribution mean. In the original implementation, L2-regularization is used instead. This may be all right for some optimizers, as for, e.g., SGD or SGD with momentum it is virtually the same as the weight decay (just rescaled by the learning rate). However, as mentioned in a paper by Loshchilov & Hutter (2017), it is not the same, for example, for ADAM optimizer, which is nonetheless exactly the one used in the original experiments. Our implementation, on the other hand, remains true to the original paper and uses proper weight decay. Nevertheless, we employ SGD with momentum in our experiments, as it demonstrated superior efficacy in our setting during our experiments.

Second, during the update of the distribution parameters, a gradient estimate should be normalized with respect to the uncertainty using a division by the standard deviation, so that it becomes a natural gradient estimate. Despite that, in the original OpenAI code, this is not done. There, this is probably hidden in the value of a learning rate (which, as a result, differs from ours). However, then the learning rate and the noise deviation are unnecessarily coupled hyperparameters.

## 3 EXPERIMENTS

We decided to test the capability of the transformers (in our case, Decision Transformers) to be trained by evolution strategies in the setting of reinforcement learning.

We chose to use the OpenAI-ES since in its core it is the simplest evolution strategy possible. It uses a simple Gaussian distribution with its parameter covariance matrix being just a $\sigma^2 \cdot I$, a rescaled identity matrix, hence no covariance is captured in this distribution, nor any difference in significance (or sensitivity) of different network weights. It does not even update the value of $\sigma$ hyperparameter during training. Therefore, if this evolution strategy works well, the others can only improve its performance.

We provided our own implementation of the OpenAI-ES and began by a replication experiment of the original paper, which also serves as a correctness check for our code. Here, we tested the algorithm on a classical feedforward network in the MuJoCo (Todorov et al., 2012) Humanoid environment using OpenAI Gym (Brockman et al., 2016), whereupon we followed up by proceeding to test the performance of the evolution strategy with the Decision Transformer in the same environment. We then examined an idea of pretraining the transformer by a behavior cloning of some smaller and easily trained – yet possibly weaker – model before training the transformer by the evolution strategy. The last experiments we present are studying the particulars of training an even larger model in a different environment with a higher-dimensional image input, which is Hero, one of the Atari games from the Arcade Learning Environment (Bellemare et al., 2013).

Because training larger models takes more time, and so training the Decision Transformer is particularly time-intensive, we chose the Humanoid environment as a representative of MuJoCo environments, since it is the most complex and challenging among the standard ones. As for the Atari, we selected the Hero game as an example of a medium-difficulty Atari game. We use Atari mainly to demonstrate training behavior on an even more complex environment with a more complex visual input, rather than to show that we can solve all its games.

In all the experiments with Decision Transformers, we used the same hyperparameter values for the model as used in the original paper (Chen et al., 2021) for the respective environments. For

---

[1]Our code, together with all the data gathered during our experiments, can be found on the following link: GitHub link (anonymized)

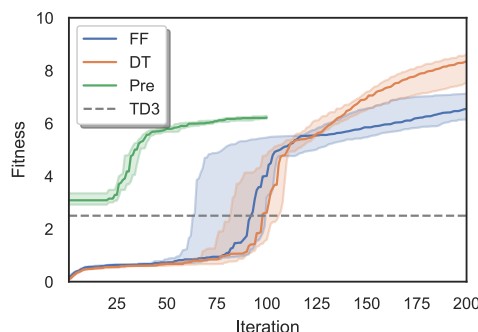

Figure 2: Median and quartiles of the best fitness values so far for all the main experiments done in the Humanoid environment for ten runs each. "*FF*" shows the training of a classical feedforward network using OpenAI-ES. "*DT*" shows the training of a Decision Transformer using OpenAI-ES. "*Pre*" shows the training of a pretrained Decision Transformer using OpenAI-ES. (This experiment differs from the previous two by the fact that its number of iterations was smaller since the model was pretrained, which is the reason why its line ends in the middle of the plot.) Finally, the "*TD3*" horizontal line shows the final average performance of the best Decision Transformer trained by TD3 in a given time.

comparison, the model sizes are 166 144 parameters for the feedforward model from the original OpenAI-ES paper (Salimans et al., 2017), 825 098 parameters for the Decision Transformer for MuJoCo Humanoid environment and 2 486 272 parameters for Atari games environment.

For all the experiments, we conducted ten runs of the training. For every run of each of the experiments, 300 workers were used utilizing 301 CPU cores (with one being a master handling synchronization, evaluation, and saving the agent). Given the computational complexity of the task and the fact that our goal is to demonstrate that the evolution is able to train the transformer, we did not wait for the run to converge. Instead, at the beginning, we assigned each run a specific number of iterations it could use to train the model. The desired returns passed to all the Decision Transformer models at the beginning of each episode were 7000 for the Humanoid Decision Transformer and 8000 for the Atari Decision Transformer. Note, that in the original Decision Transformer paper (Chen et al., 2021), for MuJoCo environments they rescale the rewards and returns-to-go dividing them by 1000. All the figures reporting fitness progression during the evolution training in Humanoid environment work with these rescaled returns as well.

Unless stated otherwise in the individual experiment descriptions, all the other hyperparameter values can be found as default values in our codebase or in Table 2 in Appendix.

To have a baseline to compare our experiments with the Decision Transformer to, we use the feedforward model trained using the OpenAI-ES during our first experiment described in Section 3.1. However, to also have some initial comparison to the gradient-based methods, we wanted to try and train the Decision Transformer using gradients. The problem with this is that the Decision Transformer is originally intended as an offline reinforcement learning model. There exists a so-called Online Decision Transformer (Zheng et al., 2022), but even it assumes starting with some pretraining. Although its architecture slightly differs from the classical Decision Transformer – it mainly differs in the action decoding layer, where a stochasticity is added – we tried to obtain the gradient baseline by training it using its standard training loop, just without any initial pretraining on precollected trajectories. However, the Online Decision Transformer proved to be unable to efficiently improve. Hence, we moved our attention to classical online reinforcement algorithms, even though those required a certain degree of creativity to make them work, since the Decision Transformer does not take only the last state as an input, but whole sequences of recent states, actions and the corresponing returns-to-go. Thus, we utilized a TD3 (Fujimoto et al., 2018), a state of the art gradient reinforcement learning algorithm, implemented by Stable Baselines3 library (Raffin et al., 2021). We ran the training for 1 500 000 timesteps – with this number of timesteps on one CPU (environment simulation) and one GPU (gradient training) it ran for roughly a similar wall-clock time as our experiments with OpenAI-ES in Section 3.2. Still, with more timesteps, the model would

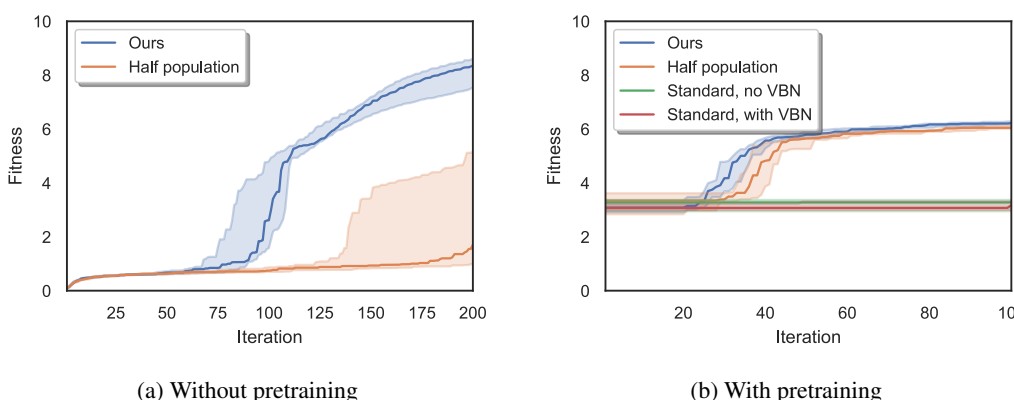

(a) Without pretraining  (b) With pretraining

Figure 3: Results of the ablation studies for training of Decision Transformers using OpenAI-ES done in the Humanoid environment for ten runs each, aggregated into medians and quartiles. Figure 3a shows the fitness progression of the best yet found model for a training without first pretraining the model. "*Ours*" stands for our setting of hyperparameters; "*Half population*" is the same except the size of the population is cut in half. Figure 3b shows the best-yet fitness values for a training utilizing pretraining. "*Ours*" and "*Half population*" are analogous to Figure 3a; "*Standard, no VBN*" has the same hyperparameters as when no pretraining was used, so the same as experiment in Section 3.2, but the virtual batch normalization is not used; "*Standard, with VBN*" has exactly the same hyperparameters as when training without the pretraining.

probably continue to improve. We conducted five runs of this gradient training in Humanoid environment and selected the final average performance of the best trained model as a baseline that is then shown in Figure 2. As we can see, the evolution is currently the only standard online reinforcement learning approach that can be applied to training the Decision Transformer without any adjustment, the gradient-based ones are either non-functional or need to be modified first.

### 3.1 FEEDFORWARD - HUMANOID

As stated before, the first experiment serves as a replication experiment for the original OpenAI-ES paper (Salimans et al., 2017), as well as a sanity check of our implementation. It consists of training a feedforward model using OpenAI-ES in Humanoid environment. The hyperparameters are thus mostly directly copied from the original paper, with the exception of those affected by our implementation changes (e.g., a learning rate) which were set manually through experiments. We can see in Figure 2 that our implementation is functional and is comparable in performance with the original implementation. The progression of the training shown in the figure also serves as a benchmark against which the results of other experiments can be compared.

### 3.2 DECISION TRANSFORMER - HUMANOID

In this experiment, we inspected the ability of the given evolution strategy to train a more complicated model. We trained a Decision Transformer model using OpenAI-ES, again in the Humanoid environment. Because the model is almost five times larger, we quadrupled the size of the population the algorithm works with. The results are shown in Figure 2. There, we can see that the algorithm is fully capable of training our larger model, achieving very good results using a constant number of iterations (albeit at the expense of a longer wall-clock runtime).

Because a larger population increases the time needed for the computation with the same available resources, a question arises whether such increase in population is truly necessary. Therefore, we also explored whether it is necessary to use such a substantial population or whether half of it would be sufficient. The results of this ablation study can be seen in Figure 3a. We can see that even though the algorithm is sometimes still capable of training the model, more often than not it does not succeed in doing so.

Last, we want to check whether the trained models respond to different returns-to-go passed to the model at the beginning of each episode as the desired performance, the target return. Hence, we simulated the trained model in the environment with various intitial return-to-go values. The results of these experiments can be found in Table 1. We can see there is no significant difference between the results of the models with distinct returns-to-go passed.

Table 1: Median and quartiles of the average (unrescaled) returns of the trained models with different initial return-to-go argument values

| Return-to-go | Q1 return | Median return | Q3 return |
|---:|---:|---:|---:|
| $-1000$ | 7466 | 8447 | 8721 |
| 0 | 7467 | 8440 | 8730 |
| 7000 | 7505 | 8370 | 8672 |
| 1 000 000 | 7489 | 8354 | 8667 |

### 3.3 Decision Transformer - Humanoid - Pretrained

The core idea behind this experiment is that if we have a really large model with many parameters, it might be quite hard for the algorithm to gain enough information from just sampling in a neighborhood of a randomly initialized model. Hence what we might try to do is first pretrain the model using behavior cloning towards some smaller, easily trained yet possibly weaker model and only then utilizing the evolution strategy to further improve the large pretrained model.

Yet, this approach has its costs. First, the OpenAI-ES uses virtual batch normalization (VBN) (Salimans et al., 2016). This changes the inputs to the model as the training progresses. So, either we would have to use VBN even during the pretraining, or we cannot use the VBN during the following training with the evolution strategy. We opted for the second possibility, as it is more general and allows us to showcase a further training of any model we might have using the evolution strategy, even though the first option would be more favorable for the algorithm, as the VBN improves its reliability (Salimans et al., 2017) – it should, however, be reportedly most important when the model is mostly random at the beginning of the training.

Second, using the same values for learning rate and noise deviation hyperparameters as when training from scratch leads to a complete randomization of the model at the beginning of the training and then the training proceeds as if from scratch. So, for the pretraining to have an effect, we have to decrease both values (to 0.01 in our experiment). This, in turn, leads to a slower progression of the training.

Hence, again in the Humanoid environment, we pretrained the Decision Transformer using a behavior cloning to a feedforward model trained using SAC algorithm (Haarnoja et al., 2018), before training it further using OpenAI-ES. The feedforward model and the pretrained Decision Transformer had a fitness value around 4. The results of the subsequent training of the pretrained Decision Transformer using the evolution strategy are shown in Figure 2. Even though our approach is capable of training the model, the progression (the improvement) is visibly slower than in the previous section without the pretraining – except for the initial phase.

Again, we explored the possibility of utilizing a smaller population for this training. As can be seen in Figure 3b, this time, although the reliability of the training is again diminished, most of the time the algorithm works surprisingly well compared to how it works with the full population. Still, there was a single run in which the model did not manage to improve.

Next, we tried using standard (i.e. the same as in the previous experiment without the pretraining) values for the learning rate and noise deviation hyperparameters both with and without the VBN, as illustrated in Figure 3b. Without the VBN the training showed no progression at all, and with the VBN the training started to show similar patterns towards the end of the training in a few cases as the one from Section 3.2, yet still was maybe a bit slower.

### 3.4 Decision Transformer - Atari

As a final experiment, we tested the ability of OpenAI-ES to train an even larger Decision Transformer in Atari games environment, more specifically, in the game *Hero*. Again, since the model is

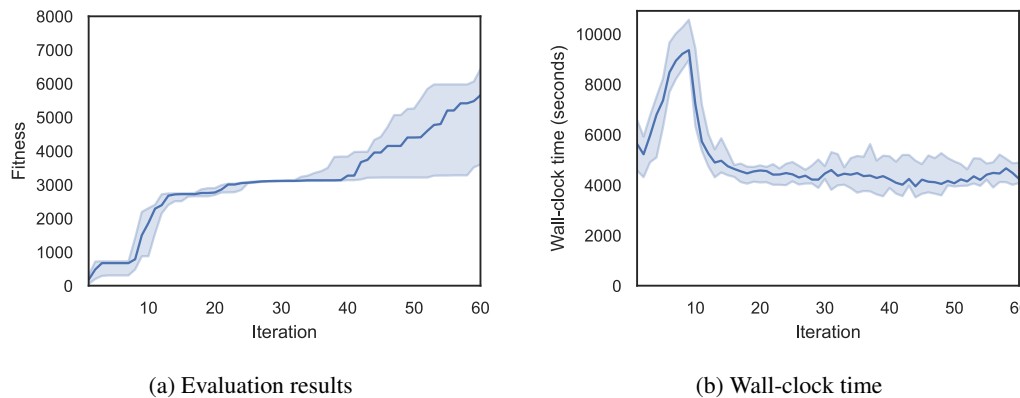

(a) Evaluation results             (b) Wall-clock time

Figure 4: Median and quartiles of fitness values and wall-clock time for OpenAI-ES used on a Decision Transformer model for Atari game Hero in ten runs of the experiment. Figure 4a shows us the best-yet evaluation results after every iteration of the algorithm. In Figure 4b, we can see the wall-clock time the training needed each iteration in seconds.

approximately three times bigger than in the previous experiments, we further doubled the population size for the algorithm. Even though Figure 4a shows that a gradual improvement of the model really does occur, Figure 4b reveals that it is accompanied by quite a significant increase in the time required to process a single iteration. This is due to a slower model inference, as the model is substantially larger, as well as longer episodes in the environment. (For the other experiments, the wall-clock time is not as interesting but can nonetheless be found alongside all the remaining collected data.) We did not include a gradient baseline for this experiment, as it primarily serves to showcase the much longer training durations required for such large models.

In Figure 4b, we can see an interesting pattern with a spike at the beginning of the training, so let us just note that it is something domain dependent. At the beginning, the agent performs random actions, which leads it to repeatedly blow itself up, hence we get relatively short episodes. Then it learns that planting the bombs randomly is not a good idea and stops ending the episodes prematurely. And then it gradually learns how to get further and further through the levels of the game, but this comes at the cost of it occasionally blowing itself up again, yet this time it at least serves its progression through the game.

## 4 DISCUSSION

As we can see by comparing the experiments in Figure 2, the larger Decision Transformer model seems to have a fitness advantage during the training over the smaller feedforward model. But what is more important, OpenAI-ES appears to be capable of training it without any substantial problems, although the process takes longer. However, the training is less robust – during one experiment run, the model failed to achieve any significant improvement. This might have been caused simply by an inconvenient seed, when the perturbations tested during the training were not suitable. Or this might be caused by one design choice of the algorithm, where for better runtime the values for perturbations are pregenerated at the beginning of the training and so for the larger models we might have to generate more values at the beginning. Or maybe we might simply need to further increase the population size for greater robustness.

Regarding our specific usage of the algorithm on Decision Transformers, we need to settle one thing. As we introduced in Section 2.2, Decision Transformer operates with return-to-go tokens, which in the original paper (Chen et al., 2021) serve as a desired return the agent should achieve in a remainder of an episode. This works well when training in the original offline reinforcement learning manner, but when using the evolution strategy to train the Decision Transformers in our online setting, there is no pressure to take these tokens into account, and thus the tokens are then ignored, as can be seen in Table 1. Nonetheless, we believe that only a minor modification of the algorithm may induce the agent to pay attention to these tokens. Let us propose an outline of such modifi-

cations for a possible future work. Each evaluation would consist of several subevaluations; each subevaluation would be assigned a desired return sampled from a $\mathcal{N}(\mu, \sigma^2)$, a normal distribution centered at $\mu$ with variance $\sigma^2$. The value of $\mu$ should be computed from the returns obtained during the previous iteration in such a manner that it is within an interval between the best return obtained and the mean return obtained during the last iteration, so an improvement is gradually made. The variance $\sigma^2$ would be a hyperparameter, or it could possibly be a variation of the returns obtained during the last iteration. The fitness of an individual would not be the mean of its returns obtained, as is the case in OpenAI-ES, but rather a decreasing function of the absolute values of the differences between the return obtained and the desired return in each subevaluation; ergo, the larger the differences, the smaller the fitness. This might be, e.g., the negative of a weighted sum of the differences.

Now, let us examine the experiments utilizing the pretraining described in Section 3.3. As we noted in the section when explaining the hyperparameter values, the pretraining does not perform very well. In order for us to be able to use the pretrained model as a base for further training – even if we would be using VBN already during the pretraining – we have to substantially reduce the ability of the algorithm to further improve the model, because we have to change the learning rate and noise deviation hyperparameters to smaller values. Hence, we get slower learning and less exploration per iteration.

Even when doing so, we can see in Figure 2 that the model first gets "broken" – its performance worsens during first few iterations, the best-yet found does not improve – before it starts to improve again. Nevertheless, the improvement comes in terms of iterations much sooner than when no pretraing is utilized (and it somewhat works even with smaller population). Furthermore, we can observe by inspecting rollouts of both the initial and final models that those two show similar gaits. This similarity gets even more accentuated, when compared to the diversity in final behaviors that are yielded from the experiment in Section 3.2, so when the algorithm is run without the pretraining. Therefore, we can see that when initialized by the pretrained model the training clearly builds upon the seeded model, despite its initial deterioration.

We hypothesize the reason for the above-mentioned deterioration at the beginning of the training from pretrained model, as well as the need for the aforementioned change of hyperparameters is the following: Gradient training – of which the behavior cloning is a representant – follows a different overall goal than the evolution strategy. It strives for the best possible parameters of the model it trains with regard to its loss function. In our case of behavior cloning, this means that we want to get our model to behave exactly as the one we are cloning towards. Nonetheless, this tells us nothing about how the model behaves when we perform a slight perturbation of parameters. Hence, after the perturbation, the model is likely to no longer function properly – or at least as well – anymore. So we can say that the model is brittle in a sense that even a slight change of parameters is likely to somehow break the model. This is, however, in stark contrast to how the evolution strategy operates and what it tries to achieve. The goal of the evolution strategy is not only that its current model performs as well as possible, but that we get the best possible behavior in expectation when sampling model parameters from its current distribution. Thus, we believe the models trained by the evolution strategy might be considered more robust – that might help, e.g., in areas where we train on precise hardware, but the model needs to be afterwards deployed on a cheaper hardware with less precision, where the weights of the model need to be rounded. Ergo, the algorithm needs to first, let us say, "robustify" the model and only then it can train it towards a better performance. This would be supported by the findings of Lehman et al. (2018).

Regardless, in the end, the results shown in Section 3.2 suggest that the pretraining is not really necessary. But then what explains why the problem stated at the beginning of Section 3.3 does not occur in practice? We think a possible answer is that the algorithm improves not only by shifting model weights towards the better performing individuals of the population, but when there are no really better individuals in the population, it does so mostly by shifting the weights away from the worst performing individuals, as the majority of the population consists of worse individuals. This might indicate why the algorithm does not really struggle with training larger and more complex models, where it is even harder to stumble onto a working solution just by adding some noise to the model parameters.

Still, it might happen in some future use-case that it would be highly beneficial to warmstart the evolution training by a pretrained model and not depend solely on the evolution to figure out everything

by itself. Our experiments hint that this should be possible to do. Let us propose a possible solution to be tested in a future work. As mentioned earlier, the currently absent VBN might be added if we use it already during the pretraining, but it should not be needed. The biggest issue with our current approach are the two lowered hyperparamer values, learning rate and noise deviation. Yes, we need to lower them for the first phase of the training, for the "robustification" of the pretrained agent, otherwise the pretrained agent gets completely broken into a random one by the evolution strategy. But after the initial phase, after we start to receive a better test results than the initial pretrained agent's ones, in theory, nothing bars us from again gradually increasing the two hyperparameters to their standard values, accelerating the training. Open question is now how fast can we increase the hyperparameters? And in Figure 3b, we can see that when pretraining we do not have to use such large population. But is this because of the two lowered hyperparameters, or because of the pretrained model? If the later would be true, it would enable us to make the pretraining even more helpful to the evolution strategy efficiency, because we could train the model using a smaller population. (And even if the former was the case, we could use a smaller population at least in the first phase of the training, still further increasing the pretraining's advantage over the training from scratch.)

Finally, let us just briefly comment on Section 3.4. The OpenAI-ES is capable of training even such larger models, but the larger the model the more computing power is required for the training to be concluded in a timely manner. Compared to sizes of transformer models powering current chatbots, the Decision Transformer used to learn the Atari game is miniscule, and even so it required a lot of wall-clock time to undertake just tens of iterations of the training on lower hundereds of CPUs. Hence, training really large transformer would require a gargantuan number of CPUs – but again, that is, at least in theory, not something unachievable. Another option is then to employ more advanced evolution strategy algorithms, possibly incorporating techniques improving efficiency. Such techniques might include utilizing a surrogate model, as can be seen, e.g., in SERL (Wang et al., 2022). This remains to be further explored.

## 5 CONCLUSION

We have investigated the ability of OpenAI-ES (and, in extension, of similar evolution strategies) to train larger and more complex models than the standard feedforward ones used in the literature so far in the reinforcement learning. We have also suggested a method to help this training by pretraining the model using behavior cloning towards some smaller, possibly weaker, yet easier to train model. However, the evolution strategy proved to be capable of training large models, whereas the pretraining showed multiple flaws during the experiments. Still, it helped us shed light onto the field of hybridizing gradients and evolution strategies, as we have discussed based on our experiments. It helped us understand why the sequential combination of first training using gradients and then proceeding to train using similar distribution-based evolution strategies might not work well. And it also allowed us to hypothesize that the distributional evolution strategies like OpenAI-ES may yield robust models, which are less prone, e.g., to rounding the weights due to low-precision hardware they might be deployed on.

### REPRODUCIBILITY STATEMENT

To ensure reproducibility of the results presented in this paper, we publish the complete codebase as well as all the data from the undertaken experiments on GitHub, which will be linked in the non-anonymized version of the paper. (The code is also enclosed in the supplementary materials to the submission, the data are too large and take too much storage space.)

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

# A  APPENDIX

Table 2: Hyperparameter values throughout the experiments (FF - Humanoid Feed Forward; DT - Humanoid Decision Transformer; Pre - Humanoid Decision Transformer Pretrained; Atari - Atari Decision Transformer)

| Hyperarameter | FF | DT | Pre | Atari | Function |
|---|---|---|---|---|---|
| rtg | N/A | 7000 | 7000 | 8000 | Return-to-go (unscaled) that should be passed to the transformer |
| size_of_population | 5000 | 20000 | 20000 | 40000 | Size of sampled population in each generation |
| num_of_iterations | 200 | 200 | 100 | 60 | Number of iterations / generations the OpenAI-ES will run for |
| noise_deviation | 0.02 | 0.02 | 0.01 | 0.02 | Standard deviation value for OpenAI-ES's distribution |
| weight_decay_factor | 0.995 | 0.995 | 0.995 | 0.995 | Decay factor of weights of the model after each update |
| batch_size | 1000 | 1000 | 1000 | 100 | Size of a batch for a batched weighted sum of noises during model update |
| update_vbn_stats_probability | 0.01 | 0.01 | 0. | 0.01 | Probability of using the data obtained during evaluation to update the Virtual Batch Normalization statistics |
| optimizer | SGDM | SGDM | SGDM | SGDM | Optimizer used in experiments |
| learning_rate | 0.05 | 0.05 | 0.01 | 0.05 | Learning rate (or step size) |

## THE USE OF LARGE LANGUAGE MODELS

The LLMs were used in a few places to improve the language of the paper and to translate several formulations to English.

