# OpenReview forum: "Utilizing Evolution Strategies to Train Transformers in Reinforcement Learning"
_ICLR.cc/2026/Conference — Submitted to ICLR 2026_

### Official Review · Reviewer_UVDc · 2025-10-15

**Soundness:** 2
**Presentation:** 2
**Contribution:** 1
**Rating:** 2
**Confidence:** 4

**Summary:**

This paper explores the use of OpenAI’s evolution strategies to train transformer-based reinforcement learning agents, specifically the Decision Transformer, in environments like MuJoCo Humanoid and Atari games. The study demonstrates that even simple evolution strategies can successfully train large and complex transformer models, though at a high computational cost. Gradient-based pretraining was tested but found to hinder rather than help performance. Overall, the work shows that evolution strategies can scale to transformer architectures, offering a robust and highly parallelizable alternative to gradient-based reinforcement learning approaches.

**Strengths:**

The key strength of this paper lies in its demonstration that evolution strategies, despite being simple and gradient-free, can effectively train large transformer-based reinforcement learning models like the Decision Transformer. This highlights the scalability, robustness, and strong parallelization potential of evolution strategies, extending their applicability to more complex neural architectures previously dominated by gradient-based methods.

**Weaknesses:**

While the paper is clearly written and experimentally careful, its novelty is quite limited. The main claim that evolution strategies can train transformer-based reinforcement learning agents is not fundamentally new. Evolution strategies have already been shown to scale effectively to large, high-dimensional models, including transformer architectures, across various domains such as natural language processing, neural architecture search, and dynamic scheduling. The present work simply extends this observation to benchmark control problems like MuJoCo and Atari, without introducing any methodological advancement or conceptual insight beyond applying existing tools to a new but predictable setting.

The section on gradient-based pretraining provides little additional contribution. The authors attempt to combine supervised pretraining with evolutionary optimization and report that pretraining hinders rather than helps performance. However, the incompatibility between gradient-trained and evolution-trained models is a well-established finding in evolutionary computation and hybrid learning literature. The paper merely reaffirms this known phenomenon without introducing a new explanation, theoretical framework, or practical remedy that would deepen understanding of how these two paradigms might be effectively integrated.

From an experimental perspective, the scope remains narrow and lacks diversity. The study tests only two standard environments, i.e., MuJoCo Humanoid and a single Atari game (Hero), which limits the generality of the conclusions. These environments, while computationally demanding, may not provide sufficiently compelling evidence that ES can outperform or even compete with modern gradient-based methods. The comparisons with baselines are weak; gradient-based methods were only minimally tuned, and the Decision Transformer’s gradient training was deemed “non-functional” without deeper investigation into stabilization techniques. As such, the experiments are more illustrative than conclusive.

Ultimately, the overall contribution of the paper is incremental. It offers an implementation and verification that ES can train transformer policies on classical control benchmarks, but this is an expected result given prior work showing ES success on transformers in other  complex domains. The paper does not deliver new algorithmic insights, hybrid methods, or theoretical understanding that could significantly advance the field. Consequently, while it may have some engineering value as a proof of concept, it falls short of making a substantive scientific contribution to reinforcement learning or evolutionary optimization research.

**Questions:**

1. How does this work advance beyond prior studies that have already applied evolution strategies to transformer architectures in other domains?

2. Why were only two standard control benchmarks (MuJoCo Humanoid and Atari Hero) used, and how do these limited experiments support broader claims about scalability or generality?

3. Given that the incompatibility between gradient-based pretraining and evolutionary optimization is well-known, what new insights or mechanisms does this paper contribute to understanding or addressing this issue?

---

> ### Author Response · Authors · 2025-11-20
>
> Thank you for your perspective on our paper. In the following, we will try to address your evaluation of our paper and try to answer the questions you raised based on your concerns throughout our answer.
>
> 1. Even though the evolution strategies (ES) have been shown to work well with transformers in various domains and this might hint that they might work even on RL domain, but before our work it has not been addressed. Beside confirming this, we also managed to lay supporting evidence to a few other claims from the literature like the difference in dynamics of distributional evolution strategies like OpenAI-ES and gradient RL algorithms [1], or providing replication experiments for the OpenAI-ES and some insights into its original implementation [2].
>
> 2. Regarding the pretraining, we initially thought that training the large Decision Transformer (DT) from a random initialization might prove too much for the ES, as described in the paper, and thus wanted to derive a feasible way of training the DT using ES. However, we were proved wrong in our experiments, since the ES was capable of training the DT from scratch. Therefore, we did not extend the idea of pretraining further. However, what we observe in the paper is not complete incompatibility, but only a poor compatibility. Still, pretraining itself does not hinder performance. The pretraining works in a sense that we are able to improve the pretrained agent by the ES, and the training performance is hindered by the fact that in the initial phase of the ES training when it gets seeded by a pretrained DT, the ES needs to "robustify" this pretrained model (seemingly breaking it in the process), as mentioned in the paper, which requires lowered learning rate and noise deviation hyperparameters (otherwise the breaking of the model is full, no longer just seeming, as can be seen in our ablations). This in turn hinders further training once this robustification phase is finished and the ES starts to further improve the model. But nothing stops us from gradually increasing these hyperparameters back to their original value once the first phase is over and we start obtaining better test results compared to the initial model performance. Plus, as mentioned in the paper, the ES training of a pretrained model is more robust in the sense that it requires smaller population (hence fewer environment episodes) to more or less reliable improve the model compared to training from scratch. This might of course be a result of lowering the two hyperparameters and maybe wouldn't hold when again increasing them as proposed above, but it might still be utilized in the initial phase of the proposed approach. This might be an interesting future work topic, thank you for bringing this detail back to our attention. We will include this in the discussion section, as it might also clarify what the takeaways from our pretraining experiment should be.
>
> 3. Concerning the experimental perspective, as mentioned in the introduction, the goal of this paper is not to compete with any SOTA algorithms. Nor is it to prove a validity of ES as a viable RL algorithm, which is shown, e.g., in the original paper introducing OpenAI-ES. [2] What we want is to show the ability of ES to train the DT. For this, the best baseline is the ES training of the simple model, i.e., our replication experiment for OpenAI-ES. The gradient "baseline" (even though it is not really a baseline in our case) was to be included only as an interesting detail. To this length, the DT’s gradient training (Online Decision Transformer) was deemed non-functional for our purposes after a few experiments for several reasons: First, it required structural changes to the DT (although minor), which defeats the purpose of the comparison. Second, even it assumes starting with some offline RL training. [3] And based on our experiments, without the pretraining, it was not able to improve the agent. (Both these resons are described in the paper.) Using the Online DT would make sense, if the ES proved to be unable to train the DT from scratch, as we would be comparing the gradient algorithm to pretrain-then-ES training pipeline, but we want a comparison to the ES training the model from scratch.

---

> > ### Author Response · Authors · 2025-11-20
> >
> > 4. The two environments were chosen as a good represetatives of their respective environment classes: The MuJoCo Humanoid is the most challenging standardly used MuJoCo environment - only exception might be the Swimmer environment, which is generally harder for the gradient algorithms, but favors the evolution, so it shouldn't be a problem in our case as well. As for the Atari Hero, it is included as an example of a more complex visual environment (compared to the simple vector states of MuJoCo) where the DT needs to be larger and also more complex (with the state encoders containing also convolution layers), hence enabling testing the dynamics of training even larger and more complex model. We specifically chose the Hero environment as an Atari representative, since it is both visually complex (multiple visually different levels) and offers a mid-level challange, but still it is not too complex like Montezuma's Revenge.
> >
> > [1] Joel Lehman, Jay Chen, Jeff Clune, and Kenneth O. Stanley. ES is more than just a traditional finite-difference approximator. In Proceedings of the Genetic and Evolutionary Computation Conference, GECCO ’18, pp. 450–457, New York, NY, USA, 2018. Association for Computing Machinery. ISBN 9781450356183.
> >
> > [2] Tim Salimans, Jonathan Ho, Xi Chen, and Ilya Sutskever. Evolution strategies as a scalable alternative to reinforcement learning. arXiv, abs/1703.03864, 2017.
> >
> > [3] Qinqing Zheng, Amy Zhang, and Aditya Grover. Online decision transformer. In Kamalika Chaudhuri, Stefanie Jegelka, Le Song, Csaba Szepesvari, Gang Niu, and Sivan Sabato (eds.), Proceedings of the 39th International Conference on Machine Learning, volume 162 of Proceedings of Machine Learning Research, pp. 27042–27059. PMLR, 17–23 Jul 2022.

---

> > > ### Comment · Reviewer_UVDc · 2025-11-22
> > >
> > > Thank you for the detailed rebuttal and for clarifying the intent behind several of your design choices. I appreciate the additional explanation regarding the expected role of pretraining and your observation that ES was ultimately able to train the Decision Transformer from scratch. However, while these clarifications help contextualize the experimental decisions, they do not substantially change my assessment of the paper’s contribution.
> > >
> > > My primary concern remains the limited novelty of the work. The rebuttal reinforces that the main contribution is confirming that ES can train transformers in RL settings, rather than providing new algorithmic insights, theoretical understanding, or unexpected empirical findings. As noted in my initial review, prior work has already established that ES scales effectively to large transformer-based models in other domains. Simply demonstrating that this behavior extends to MuJoCo and Atari constitutes an incremental extension rather than a substantive advance. The additional points raised in the rebuttal, such as replication experiments and implementation observations, are appreciated but do not elevate the scientific contribution beyond empirical validation.
> > >
> > > Regarding the pretraining component, the rebuttal mentioned the “robustification” phase and potential future approaches such as adaptive hyperparameters fine-tuning. However, this remains speculative and does not provide new insight into the well-documented challenges of combining gradient-based training with evolutionary optimization. In my opinion, the paper still lacks a deeper analytical or methodological contribution that would help the community understand or overcome this incompatibility.
> > >
> > > Finally, the rebuttal does not fully address the concerns about experimental scope and baseline strength. While I acknowledge the computational constraints and the rationale for selecting Humanoid and Hero, the narrow domain coverage and lack of competitive gradient-based baselines mean that the claims about scalability and generality remain insufficiently supported, limiting the overall impact of the findings.

---

### Official Review · Reviewer_7Pvx · 2025-10-26

**Soundness:** 2
**Presentation:** 3
**Contribution:** 3
**Rating:** 4
**Confidence:** 3

**Summary:**

This paper covers an interesting idea of using evolutionary algorithms to train transformer-based RL agents. Using OpenAI-ES, the algorithm was tested on various Atari and a Mujoco environment, showing improved performance over TD3. The paper also includes an interesting discussion and insights on the strengths and weakness of evolution strategies and their possible future implications in the field of RL.

**Strengths:**

This paper covers an interesting question of using evolutionary strategies to train transformer architectures in an RL setup. Given the recent interest in transformer architecture, whether in decision making (Decision Transformer) or reasoning (LLMs and Reasoning Models), this sheds lights into an interesting research direction. This paper also includes some promising results and insights, providing a useful starting point for future research into evolutionary strategies for transformer architectures.

The paper is also very well organized. It was quite easy and enjoyable to read the paper.

**Weaknesses:**

While this paper offers a promising vantage point into future research, I think there are several improvements that can be made to offer more to the RL community.

First, while the experiment only includes a dotted-line comparison with TD3, I think it would be better to include a more though comparison with more RL algorithms (PPO, SAC, etc) as well as their resource usage plots. While evolutionary strategies often consume a lot more computational resource than more traditional RL algorithms, many of their process can easily be parallelized and with the advent of JAX, this approach is much more accessible. Therefore, a comparison with existing RL algorithm in terms of computational resources used and time elapsed would present a more useful information for future researchers whether genetic algorithms would be applicable to their research topics.

Related to above, including tasks other than Mujoco and Atari would be more useful. Various RL tasks, from autonomous driving to GRPO used for LLMs, has their own unique challenges and purposes. Providing a more detailed understanding on where the evolutionary strategy works and have bonus over existing RL algorithms would provide a useful markers for future researchers whether they should use an evolutionary strategy or not.

The fact that pretraining does not help seems like a critical issue, as there are some tasks where the dataset is not big enough or the model is too big that it would be impossible to train the agent without pretraining on a similar and more simple set of tasks. Therefore, a comprehensive experiments on where and why pretraining works or not works would be useful.

While the discussions were very clear, honest, and shared many useful insights about the approach, there were some cases where the claims were not well supported and simply marked 'We think'. While such clarity is certainly welcomed, I think it would be nice to conduct some ablation tasks to validate theories presented in the discussions.

**Questions:**

Overall, this is a promising research and for me, the biggest factors for improving score would be

1) comparison in resource usage for genetic strategies compared to regular RL algorithms
2) more ablation studies to support the thoughts and claims made in the discussion.

---

> ### Author Response · Authors · 2025-11-20
>
> Thank you for the interesting points you raised. We will try to react to your concerns and address possible misunderstandings.
>
> 1. In the original paper introducing the OpenAI-ES algorithm [1], the authors trained simple feed-forward network-based agent on MuJoCo Humanoid in 10 minutes on 1440 CPUs. So the parallelizability is indeed great for this evolution strategy (ES). We constrained ourselves to a 300 workers for each run, as mentioned in our paper, so the wall-clock time is not as great as in the mentioned work, but further parallelization onto more CPUs would easily improve it. Nonetheless, for those further interested, the data regarding wall-clock time of all our experiments are contained in the logs, which will be available on the link contained in the deanonymized version of our paper.
>
> 2. The problem with other gradient algorithms is they are not out-of-the-box prepared to train the Decision Transformer (DT). Input to the DT is not only the last state, but also the whole sequence of several states, actions taken, and returns-to-go. Thus to train it using the classical gradient algorithms without much change to them, we have to save all these sequences whole as "states" for each transition, especially if the algorithm uses replay buffer from which the transitions are sampled. This increases memory requirements (even more so for environment with already complex states) and changes the MDP being solved. Still, the comparison of gradient and evolutionary algorithms is out of the scope for this work, even though it might be found in a broader literature. [2,3] For our purpose, which is showing the ability of ES to train the DT, we consider the ES-trained simple model to be much better and more telling baseline, and include the gradient training for completeness and as a matter of interest. This is also the reason for not including other environments. We want to show that ES is capable of training DT. The ability of ES to train a simple agent in various environments and the fact it is a viable alternative to gradient algorithms (with various pros and cons) are generally accepted by the evolutionary policy search community. [2,3,4]
>
> 3. As for your note regarding the pretraining, we also initially thought the model might prove too big and complex to be trained from scratch. However, as discussed in the paper, we were proved wrong on this point by our experiments and the only cost for the larger model was larger population of the ES, meaning more environment episodes per generation - which can in theory be parallelized out of relevance. This then resulted in the same number of generations needed for the large model as for the simple one. That is also why we did not investigate the pretraining further - even without it, the algorithm works quite well. However, we can see that the ES is capable of building on the pretrained model, although this has the cost in the form of lowered "learning speed" hyperparameters, the learning rate and noise deviation, as mentioned in the discussion, and that the improvement of the pretrained model comes much sooner than that of the randomly initialized one despite the lower hyperparameters. But nothing bars us from gradually increasing these hyperparameters to the original values once the test performance improves over the initial pretrained model's performance, thus regaining the speed of the training. The only problem remaining would be the absence of VBN, but the VBN should help mostly in the beginnings of ES training according to the original authors of OpenAI-ES [1], so it shouldn't hurt our pretrained model-seeded training. Or we could easily solve the VBN's absence by using the VBN already during the pretraining, which would allow us to use it even in ES training. We will include this observation in the discussion to further enhance the paper and as a hint of a possible future work in the area. Thank you for raising this point, we believe that adding this may further increase the value of our paper's contribution.
>
> 4. And finally, in the discussion, there are three main points we hypothesize about: the possibility of forcing the ES-trained DT to pay attention to the return-to-go tokens, the dynamics of combination of ES and gradients, and why are we able to gain enough information to improve from just sampling in a neighborhood of a large randomly initialized model when an interesting (well rewarded) behavior is unlikely to occur randomly. The first point is just a possible direction for a future work we wanted to include for those interested. The second hyphotesis is supported by [5], as stated in our paper. And the third we did not plan to examine more closer, but seeing it might be of interest, we may look into it in a future work and agree that it might be informative to disect it more - thank you for pointing out that it might be interesting.

---

> > ### Author Response · Authors · 2025-11-20
> >
> > [1] Tim Salimans, Jonathan Ho, Xi Chen, and Ilya Sutskever. Evolution strategies as a scalable alternative to reinforcement learning. arXiv, abs/1703.03864, 2017.
> >
> > [2] Amjad Yousef Majid, Serge Saaybi, Vincent Francois-Lavet, R. Venkatesha Prasad, and Chris Verhoeven. Deep reinforcement learning versus evolution strategies: A comparative survey. IEEE Transactions on Neural Networks and Learning Systems, 35(9):11939–11957, 2024.
> >
> > [3] Olivier Sigaud. Combining evolution and deep reinforcement learning for policy search: A survey. ACM Trans. Evol. Learn. Optim., 3(3), September 2023.
> >
> > [4] Yang YU, Hong QIAN. Derivative-free reinforcement learning: a review. Frontiers of Computer Science, 15(6):156336, 2021.
> >
> > [5] Joel Lehman, Jay Chen, Jeff Clune, and Kenneth O. Stanley. ES is more than just a traditional finite-difference approximator. In Proceedings of the Genetic and Evolutionary Computation Conference, GECCO ’18, pp. 450–457, New York, NY, USA, 2018. Association for Computing Machinery. ISBN 9781450356183.

---

> > > ### Comment · Reviewer_7Pvx · 2025-11-24
> > >
> > > Thank you for your comments. Rebuttals 1,3,4 seems sound, however, I would like to hear more about Rebuttal 2.
> > >
> > > There are several ways in which a decision transformer can be trained. For example, there are several notable works in which a decision transformer was trained offline with reward conditioned supervised learning. While suggesting a new research direction in to using evolutionary strategies is certainly intriguing, I think potential reader would find it much more beneficial if there were experiment results and analysis where and when should we use the proposed algorithm to train a decision transformer, instead of using more established works. I think such comparison would make it much better to justify the novelty of the work.

---

> > > > ### Author Response · Authors · 2025-11-26
> > > >
> > > > As you mentioned, Decision Transformer has originally been introduced as an offline RL model [1] with its training constituting of the reward conditioned supervised training on precollected trajectories. Even the Online Decision Transformer (ODT) we mention in the paper assumes starting with some offline RL training first, before continuing to train in online regime. [2] And based on our experiments, without the pretraining, it was not able to improve the agent. (Plus it requires slight structural changes to the model.) We also quickly checked the newest articles regarding the DT and all the proposed approaches build on the ODT and improve its online finetuning, so the offline part remains in place.
> > > >
> > > > Compared to this, training the DT using the evolution is fully online, when we do not use the pretraining. And even with the pretraining, there would be a difference: The offline RL assumes dataset of diverse trajectories on which the agent is trained. Our pretraining is a simple behaviour cloning towards a single (online trained) agent, which could generate just slight variations of the same trajectory, which would not be sufficient for successful offline RL.
> > > >
> > > > We have now included this distinction between the evolutionary approach we test and the offline / offline-to-online gradient training in the introduction of our paper, thank you.
> > > >
> > > > [1] Lili Chen, Kevin Lu, Aravind Rajeswaran, Kimin Lee, Aditya Grover, Michael Laskin, Pieter Abbeel, Aravind Srinivas, and Igor Mordatch. Decision transformer: Reinforcement learning via sequence modeling. arXiv preprint arXiv:2106.01345, 2021. doi: 10.48550/arXiv.2106.01345.
> > > >
> > > > [2] Qinqing Zheng, Amy Zhang, and Aditya Grover. Online decision transformer. In Kamalika Chaudhuri, Stefanie Jegelka, Le Song, Csaba Szepesvari, Gang Niu, and Sivan Sabato (eds.), Proceedings of the 39th International Conference on Machine Learning, volume 162 of Proceedings of Machine Learning Research, pp. 27042–27059. PMLR, 17–23 Jul 2022.

---

### Official Review · Reviewer_RjaQ · 2025-10-31

**Soundness:** 3
**Presentation:** 4
**Contribution:** 3
**Rating:** 8
**Confidence:** 4

**Summary:**

The paper studies whether Evolution Strategies (ES) can train transformer-based RL agents such as Decision Transformers. The authors test this on MuJoCo Humanoid and the Atari game HERO. They also try training a model after behavior-cloning pretraining. The main claim is that ES can scale to transformer models and that ES-trained policies tend to become robust during training.

**Strengths:**

The paper is very relevant.

The motivation is clear. The authors want to understand if ES can handle modern, large RL architectures. This is timely and relevant.

The experimental setup shows solid engineering effort. They re-implemented OpenAI-ES and carefully described the main design decisions.

The paper presents an interesting observation. ES initially weakens a pretrained model before improving it. The authors argue that ES first improves robustness in parameter space. This is a useful insight for hybrid ES + gradient training.

The results show that ES can train a Decision Transformer from scratch on a continuous-control task. This is non-trivial and demonstrates feasibility.

**Weaknesses:**

The contribution feels more like a feasibility study than a new method or strong theoretical insight. The paper could benefit from clearer framing about what new knowledge is gained beyond “ES works on transformers.”

The baseline using TD3 with a Decision-Transformer-style model is not strong. It mostly shows that standard RL fails here. Stronger or more relevant baselines (e.g., modern online sequence models or RvS approaches) would help.

The return-to-go signal essentially gets ignored when using ES. The paper mentions this but does not analyze the reason deeply. Some diagnostic experiments or further discussion would strengthen the argument.

Atari results only include one game (HERO). More environments would make the general claims more convincing.

ES is known to be compute-heavy. The paper shows runtime, but there is little discussion of compute vs. performance or sample efficiency. This would help place the results in context.

**Questions:**

Can you provide results for more Atari games? There has, after all, been more than a month since you submitted your paper.

Can you provide more info on sample efficiency?

---

> ### Author Response · Authors · 2025-11-20
>
> Thank you for your review, in the following we will try to address the concerns you expressed and hopefuly answer your questions.
>
> 1. We agree that this paper is mainly a feasibility study of training Decision Transformers (DT) via Evolution strategies (ES), however it also contains several other insights (or confirmations of previous works) throughout its discussion:
>     - The comments on computational requirements or observations concerning dynamics of utilizing the pretraining regarding the small compatibility based on a different optimization objective of gradients and evolution (confirming previous work [1]).
>     - The fact that despite this poor compatibility, the evolution seeded by the agent pretrained by gradient retains the pretrained walking gait, even though it has to first completely dismantle the agent's behavior to robustify the population distribution (a new insight).
>     - Also, the paper doubles as a replication study for the OpenAI-ES. [2]
>
> 2. We included the gradient training not as much as a performance baseline - for that, we believe, the ES-trained simple model is more suitable, since it uses the same hardware and other resources - but just as an example of a gradient general online RL algorithm used to train the DT. As we understand it, the RL via Supervised Learning (RvS) is an offline RL approach while training using the ES is a fully online RL approach (when no pretraining is utilized), so they tackle different challenges. (Similar problem holds also for the Online Decision Transformer mentioned in the work, which should keep improving the model in online manner, but it requires first some offline pretraining first in order to work properly - without the pretraining, the model was incapable of improving itself in our experiments.) Concerning the "modern online sequence models", we are not sure which models to consider here, we were able to find only approaches that are hardly applicable to our topic (Multiagent RL, PPO derivatives for LLMs, ...). Any further advice would be welcomed for a future work.
>
> 3. Ignoring of the return-to-go signal by the DT is surely an interesting point which, as we mention in the discussion, where we propose a possible approach to force the ES-trained DT to pay attention to the signal, remains for a future work.
>
> 4. We understand that when training on Atari, it is customary to train on all, or at least a substantial portion of the games. But we believe that other tested Atari environments wouldn't bring that much to the paper. The Atari Hero is included only as an example of a more complex visual environment - compared to the MuJoCo, where the states are simple vectors - where the DT needs to be larger and also more "complex" (with the state encoders containing also some convolution layers), hence enabling testing the dynamics of training even larger and more complex model. We specifically chose the Hero environment as an Atari representative, since it is both visually complex and offers a mid-level challenge - there are several stages of multiple levels, each level has a different background color, also sometimes the player has to choose from various paths, some of the actions have delayed effect, e.g., the usage of a bomb, which is even possibly deadly despite it is vital that the player uses them - still it is not too complex like Montezuma's Revenge.
>
> 5. Regarding the sample efficiency, we use an existing and in this regard explored algorithm belonging to the family of evolutionary algorithms, which is thoroughly examined in a broader literature. [3,4] Generally, the evolutionary algorithms have poorer sample efficiency than the gradient algorithms. The evolution samples multiple full episodes and then proceeds with one update to the agent, while the gradients (and more so the Temporal Difference-like methods) squeeze as much information as possible from each step in the environment and use it for several updates. However, the evolution offers easier parallelization, which might result in a better wall-clock time.
>
> [1] Joel Lehman, Jay Chen, Jeff Clune, and Kenneth O. Stanley. ES is more than just a traditional finite-difference approximator. In Proceedings of the Genetic and Evolutionary Computation Conference, GECCO ’18, pp. 450–457, New York, NY, USA, 2018. Association for Computing Machinery. ISBN 9781450356183.
>
> [2] Tim Salimans, Jonathan Ho, Xi Chen, and Ilya Sutskever. Evolution strategies as a scalable alternative to reinforcement learning. arXiv, abs/1703.03864, 2017.
>
> [3] Amjad Yousef Majid, Serge Saaybi, Vincent Francois-Lavet, R. Venkatesha Prasad, and Chris Verhoeven. Deep reinforcement learning versus evolution strategies: A comparative survey. IEEE Transactions on Neural Networks and Learning Systems, 35(9):11939–11957, 2024.
>
> [4] Olivier Sigaud. Combining evolution and deep reinforcement learning for policy search: A survey. ACM Trans. Evol. Learn. Optim., 3(3), September 2023.

---

### Meta-Review · Area_Chair_Pgzb · 2026-01-06

**Summary:**

This paper proposes the use of evolutionary strategies to train transformer architectures for reinforcement learning. The authors' main point seems to be to argue that evolutionary strategies _can_ be used to train transformer architectures, which they do provide evidence for. However, the paper reads more as an initial feasibility study (as stated by the authors themselves, in response to RjaQ), but would require a lot more depth in empirical evaluation before it is ready for acceptance. In their response to RjaQ, the authors argue some of the novel insights their work provides; while interesting, they are not strong enough for me to recommend acceptance.

See below for the main reviewer concerns which I felt were left unaddressed.

**Reviewer Concerns:**

Most of the main concerns were _not_ properly addressed by the authors. Notably:

## RjaQ
- Not convincing when arguing for the submission being more than a feasibility study
- When requested to run on more Atari games, the authors' response was "other tested Atari environments wouldn't bring that much to the paper", which is simply not true.
- When asked about sample efficiency, the authors claim ES offers easier parallelization, which is an unsubstantiated claim. Indeed, there have been a number of gradient-based methods which use parallelization, such as IMPALA.

## 7Pvx
- Requested wall-clock time, which the authors replied by saying they will be available in the logs in the deanonymized version (not clear why they couldn't share it now)
- No proper comparison with other RL algorithms which do lend themselves to parallelization.
- When probed about the ineffectiveness of pre-training, the response was a bit of a non-answer: the authors argued that because it worked well without it they didn't feel it was necessary to investigate further.
- When probed about the speculative statements (lots of 'We think's), the authors simply state that they are points for future work. A number of these could have been investigated more concretely during the rebuttal (for instance with regards to pretraining as well as with their claim that ES first improves robustness in parameter space).

## UVDc
- Narrow experimental scope and diversity, which the authors did not address directly ("the experiments are more illustrative than conclusive").
- With regards to pretraining, the claim that "ES needs to 'robustify' this pretrained model" is unsubstantiated and speculative.
- When discussing hyperparameter selection, the authors suggest there may be room for more exploration but fail to provide these analyses ("This might of course be a result of lowering the two hyperparameters and maybe wouldn't hold when again increasing them as proposed").
- The reviewers end with the following statements, which suggest the paper is not ready for publication:
  - "The rebuttal reinforces that the main contribution is confirming that ES can train transformers in RL settings, rather than providing new algorithmic insights, theoretical understanding, or unexpected empirical findings."
  - "this remains speculative and does not provide new insight into the well-documented challenges of combining gradient-based training with evolutionary optimization. In my opinion, the paper still lacks a deeper analytical or methodological contribution that would help the community understand or overcome this incompatibility."

**Reviewer Scores:**

- **RjaQ:** currently 8, likely to remain at 8
- ** 7Pvx:** currently at 4, unlikely to increase
- ** UVDc:** currently at 2, unlikely to increase

---

### Decision · Program_Chairs · 2026-01-26

Reject